# Ultrastructure of macromolecular assemblies contributing to bacterial spore resistance revealed by in situ cryo-electron tomography

Elda Bauda [1], Benoit Gallet [1], Jana Moravcova[2], Gregory Effantin [1], Helena Chan[3], Jiri Novacek [2], Pierre-Henri Jouneau [4], Christopher D. A. Rodrigues [5], Guy Schoehn [1], Christine Moriscot [6,7] ✉ & Cecile Morlot [1,7] ✉

Bacterial spores owe their incredible resistance capacities to molecular structures that protect the cell content from external aggressions. Among the determinants of resistance are the quaternary structure of the chromosome and an extracellular shell made of proteinaceous layers (the coat), the assembly of which remains poorly understood. Here, in situ cryo-electron tomography on lamellae generated by cryo-focused ion beam micromachining provides insights into the ultrastructural organization of *Bacillus subtilis* sporangia. The reconstructed tomograms reveal that early during sporulation, the chromosome in the forespore adopts a toroidal structure harboring 5.5-nm thick fibers. At the same stage, coat proteins at the surface of the forespore form a stack of amorphous or structured layers with distinct electron density, dimensions and organization. By analyzing mutant strains using cryo-electron tomography and transmission electron microscopy on resin sections, we distinguish seven nascent coat regions with different molecular properties, and propose a model for the contribution of coat morphogenetic proteins.

Sporulation is a complex developmental process mainly found in Gram-positive bacteria (Bacilli and Clostridia), leading to a dormant cell type (the spore) resistant to most of the treatments used to eliminate vegetative bacteria: antibiotics, high temperatures, detergents and irradiation[1]. This toughness results from the assembly of durable intracellular and extracellular structures that protect the cell and its genetic material, while keeping it receptive to its environment so that it can germinate and resume vegetative growth under appropriate conditions.

Upon environmental stress, such as nutrient deprivation, spore-formers enter a differentiation process, starting with an asymmetric division that results in two morphologically different compartments: a large mother cell and a small forespore (Supplementary Fig. 1, stage I)[2]. These two cells, which form the sporangium, are genetically identical but they follow specific gene expression programs, governed by sporulation-specific sigma factors (σ) whose timely activation is ensured by molecular checkpoints taking place at the mother cell and forespore interface[2,3]. σF and σG respectively control early and late

[1]Univ. Grenoble Alpes, CNRS, CEA, IBS, F-38000 Grenoble, France. [2]CEITEC-Central European Institute of Technology, Masaryk University, 62500 Brno, Czech Republic. [3]University of Technology Sydney, 2007 Ultimo, NSW, Australia. [4]University Grenoble Alpes, CEA, IRIG-MEM, F-38054 Grenoble, France. [5]School of Life Sciences, University of Warwick, Coventry, UK. [6]Univ. Grenoble Alpes, CNRS, CEA, EMBL, ISBG, F-38000 Grenoble, France. [7]These authors jointly supervised this work: Christine Moriscot and Cecile Morlot. ✉e-mail: christine.moriscot@ibs.fr; cecile.morlot@ibs.fr

gene expression in the forespore, while σE and σK respectively drive early and late expression of mother-cell specific genes. Following asymmetric division, the two cells are separated by the so-called intermembrane space (IMS), composed of two membranes and septal peptidoglycan (PG). Septum thinning is then observed while the chromosome is pumped into the forespore by the SpoIIIE ATPase (Supplementary Fig. 1, stage $II_E$)[4,5]. At this stage, the mother cell membrane starts migrating around the forespore, through a process called engulfment (Supplementary Fig. 1, stage $II_M$), which requires coordinated PG synthesis and degradation[6,7]. The engulfed forespore is eventually surrounded by its own cytoplasmic membrane (called the inner forespore membrane, IFM) and a second membrane derived from the mother cell (called the outer forespore membrane, OFM) (Supplementary Fig. 1, stage III). Tethering of these two membranes involves the assembly of a transenvelope nanomachine called the SpoIIIA-SpoIIQ complex[6,8]. Halfway through engulfment, protective protein layers (called the coat) begin to self-assemble at the surface of the OFM[9]. At the end of engulfment, a modified PG, called the cortex, is synthesized in the IMS (Supplementary Fig. 1, stage V)[10]. Eventually in *Bacillus subtilis*, assembly and maturation of the coat layers will form a thick protective envelope shield, divided into four sub-layers: the basement layer, the inner coat, the outer coat and the crust[9,11] (Supplementary Fig. 1, stage VII).

Despite their importance for the acquisition of resistance properties, the mechanisms involved in the maturation of the forespore are not yet fully elucidated, mainly because they involve molecular multiprotein complexes of nanometric dimensions, whose assembly usually requires the cellular environment. A first example is the conformational change undergone by the chromosome after its transfer into the forespore. It transitions from a compact structure to a toroid, and eventually adopts a crystalline arrangement (Supplementary Fig. 1, stage VI-VII)[12]. This sequence of events cannot be reproduced in vitro and has therefore been little documented. The spore coat is also poorly understood, as this composite structure can only assemble in vivo, from a complex network of interactions involving first a tens morphogenetic proteins, and eventually >80 different proteins[9]. SpoVM, SpoIVA, SpoVID, SafA and CotE are the main morphogenetic proteins, as their absence causes major defects in early coat architecture. Above the OFM, SpoVM and SpoIVA form the basement layer, whose migration around the circumference of the forespore (encasement) is assisted by SpoVID. SafA and CotE are respectively required for the deposition of the inner and outer coat. Eventually, layers of different compositions and properties will be built from SpoVM-SpoIVA, SpoVID, SafA and CotE nursery structures whose architecture remains uncharacterized. A major challenge in the structural study of the spore coat is the poor solubility of coat proteins when produced as recombinant constructs, and the extensive cross-linking within coat layers that prevents their purification from spores. Investigation of such molecular processes thus requires high-resolution in situ observation methods. Cellular electron microscopy (EM) meets this need, but its resolution capacity is limited to specimens that are generally thinner than 300 nm. In most cases, bacterial cells are too thick to be observed with high resolution and contrast. In the last decade, focused ion beam micromachining monitored by scanning electron microscopy at cryogenic temperature (cryo-FIBM/SEM) emerged as a powerful method for thinning the specimen[13–16]. From a cell layer vitrified at the surface of an EM grid, lamellae of 150-200-nm thickness are milled by a gallium beam, which gives access to deep cell regions with minimal milling artifacts.

In this work, we combine cryo-FIBM with cryo-electron tomography (cryo-FIBM/ET) to observe *B. subtilis* sporangia during early- (stage $II_E$), mid-engulfment (stage $II_M$) and post-engulfment stages (stage III) (Supplementary Fig. 1). The high contrast in the reconstructed cryo-ET data (Supplementary Movies 1,2) allows segmentation of various mother cell and forespore ultrastructures

(Figs. 1–3, Supplementary Fig. 2 and Supplementary Movies 3,4). This analysis reveals early stages of DNA packing inside the forespore and the complex organization of nascent coat layers at the surface of the OFM. Key mutant strains, analyzed by cryo-FIBM/ET or by transmission electron microscopy (TEM) on resin sections of freeze-substituted *B. subtilis* sporangia, provide clues to identify the nature of these layers.

## Results

### Ultrastructures of the mother cell and forespore envelopes

Following optimization of the cryo-FIBM workflow for *B. subtilis* (Supplementary Methods), cryo-ET data were collected at high defocus (−10 to −20 μm) and medium magnification (pixel size of ~4.5 Å) to visualize large area of the cell with high contrast, which facilitated segmentation (Figs. 1–3, Supplementary Fig. 2 and Supplementary Movies 3,4). Three *B. subtilis* strains were studied by cryo-FIBM/ET. We first used a *ΔspoIVB* strain, in which impaired σK activation blocks sporulation at stage IV[3], to reduce the diversity of proteins composing the coat layers to σE-dependent ones. We also acquired tilt series on *ΔspoIIQ* and *ΔcotE* strains, which display phenotypes that were key to investigate coat assembly[17,18].

The homogeneous shape of the cells in our tomograms, as well as the integrity of the membranes and of membrane complexes, like chemotaxis arrays, illustrate good preservation of our samples (Fig. 1a and Supplementary Fig. 2). Our protocol even allowed preserving fragile surface complexes such as flagella or heterogeneous structures resembling flowers on stems, whose nature remains to be identified (Supplementary Fig. 2). The high contrast in our tomograms reveals that cells are covered by sharp densities of $17.6 \pm 1.2$ nm ($n = 24$) in height (Fig. 1b). Upon segmentation, these objects appear distributed as lines, oriented along the cell circumference and separated by ~12 nm (Fig. 1c).

The homogeneous density of the mother-cell PG layer also testifies to the good preservation of the samples (Fig. 1b). Consistent with previous observations[5], its thickness in longitudinal cell sections is $26.9 \pm 1.3$ nm ($n = 36$) (Supplementary Table 2). Interestingly, in many post-engulfment cells ($n_{ΔspoIVB} = 8$ out of 15, $n_{ΔcotE} = 19$ out of 40), the envelope of the mother cell bulges towards the forespore at positions close to the sporangium pole (Fig. 1d). These prominences cannot correspond to the junction between the vegetative PG and the septal PG synthesized during asymmetric division because they do not encircle the mother cell. In addition, they are not visible in engulfing sporangia; they are therefore rather the result of a post-engulfment event.

We observe two thin dark lines parallel to the cytoplasmic membrane in the mother cell wall of our thinnest cryo-FIBM sections, including in the bulging area (Fig. 1e). These lines, made of dense material, are either close to the membrane or located in the middle of the PG layer. In engulfing regions, thin dark lines are similarly observed above the mother cell cytoplasmic membrane, ahead and behind the engulfing front (Fig. 1f). In the IMS, thin lines can also be observed in thin sections (Fig. 1f), but they are not resolved in thick ones (Fig. 1b). Thick sections may not only contain more of these linear objects but also present a lower signal-to-noise ratio. Consequently, if these lines do not overlap perfectly in the plane perpendicular to the lamella, they may result in a larger structure within which the individual lines cannot be resolved. This dense material at the center of the IMS, which was previously proposed to be a thin layer of PG[5,19], has a thickness of about $6.0 \pm 0.5$ nm ($n = 24$, calculated as the thickness of the single dark line or as the distance separating the two thin lines). Although the nature of the thin dark lines observed in the mother cell wall (Fig. 1e) remains to be clearly established, the membrane-proximal ones might correspond to nascent glycan strands still attached to the cytoplasmic membrane, while the more distant ones might correspond to glycan strands that are being incorporated into the PG network.

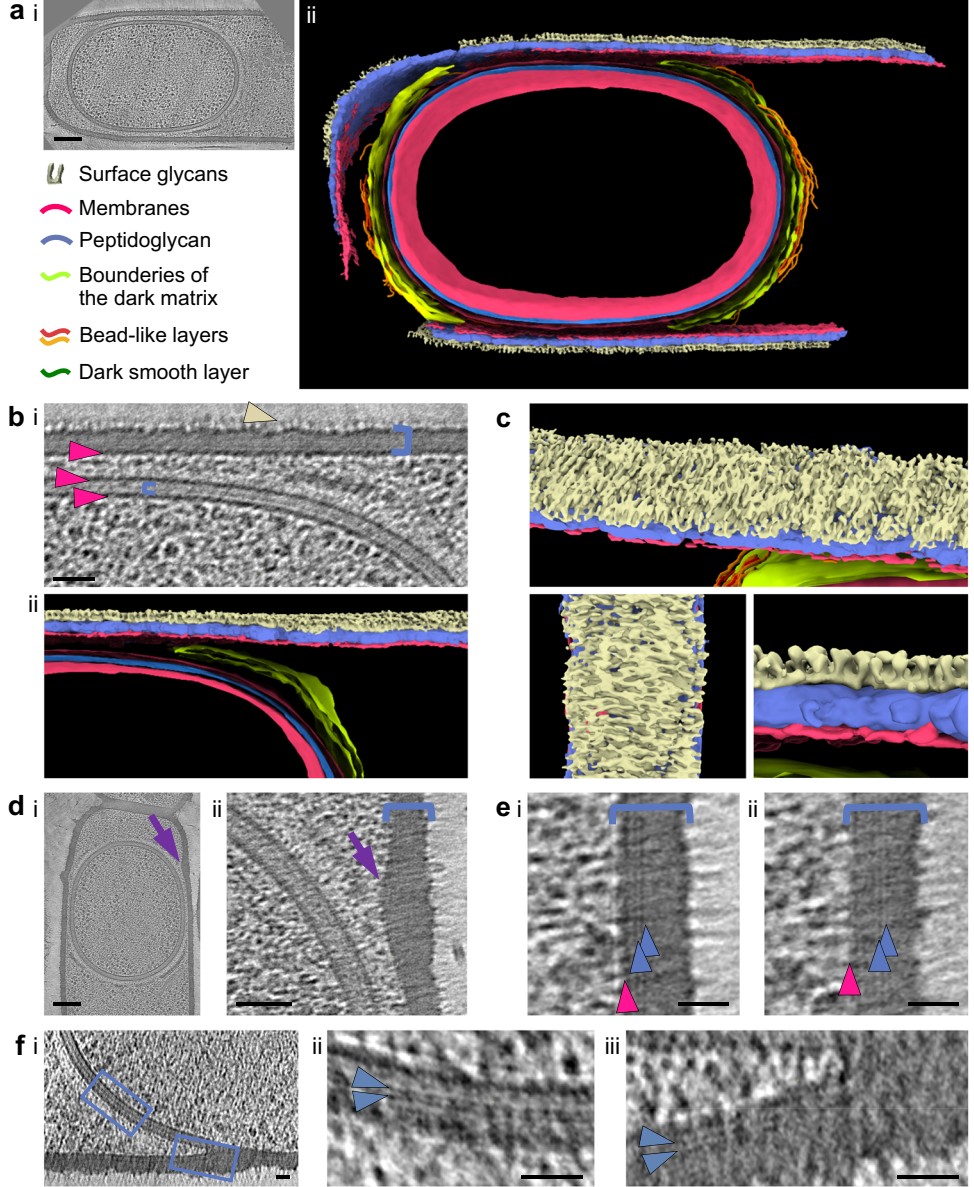

**Fig. 1 | Cellular ultrastructures of *B. subtilis* sporangia. a** Slice through a tomogram (**i**) of a stage-III *ΔspoIVB B. subtilis* sporangium, used for the segmentation (**ii**) of various forespore and mother cell ultrastructures. The image is representative of 2 independent experiments, with 15 cells displaying similar features. Scale bar = 100 nm. **b** Zoom of a slice through a tomogram (**i**) showing the spore PG layer (small blue bracket) sandwiched between the IFM and OFM (lower magenta arrowheads), as well as the mother cell membrane (top magenta arrowhead), PG layer (large blue bracket) and surface glycans (light yellow arrowhead). Scale bar = 50 nm. Panel **ii** shows the corresponding segmentation. **c** Different views of the segmented mother cell envelope showing surface glycans. **d** Full view (**i**, scale bar = 100 nm) and zoom (**ii**, scale bar = 50 nm) of a slice through a cryo-electron tomogram in which the mother cell envelope bulges (violet arrow) toward the forespore. The images are representative of 2 independent experiments, with 27 cells displaying similar features. **e** Zooms of tomogram slices showing two thin dark lines (blue arrowheads) located close (**i**) to the mother cell cytoplasmic membrane (magenta arrowhead) or in the middle (**ii**) of the PG layer (blue bracket). Scale bars = 20 nm. **f** Large view (**i**) and zooms (blue insets) of a tomogram slice showing the region of the engulfing front, in which two thin dark lines (blue arrowheads) are observed in the middle of the IMS (**ii**) or close to the mother cell cytoplasmic membrane (**iii**). Scale bars = 20 nm.

## Cytoplasmic and surface ultrastructures of the forespore

The cytoplasmic content of the sporangia was analyzed in *ΔspoIVB* and *ΔcotE* sporangia, in which forespores have a regular, round to oblong shape, as observed in the wild-type strain[5]. Inside the forespore, ribosomes were easily identified in all sporangia ($n_{ΔspoIVB}$ = 25, $n_{ΔcotE}$ = 58) (Supplementary Fig. 2a) and DNA was visible in thin sections of stage-III forespores ($n_{ΔspoIVB}$ = 15, $n_{ΔcotE}$ = 40) (Fig. 2). Interestingly, in 23 of these tomograms, the DNA displays a fibrous appearance, with bundles of about 5.5 ± 0.8 nm ($n$ = 36) in diameter. Furthermore, the chromosome forms a crescent- or a toroid-shaped structure around a DNA-free region (Fig. 2a, b). In 10 other forespores, we distinguished

two distinct DNA-rich regions (Fig. 2c), which are consistent with sections orthogonal to that of Fig. 2b. Altogether, these observations support the idea that the chromosome in engulfed forespores adopts a toroidal shape.

The high signal-to-noise ratio of our images provided exquisite details of the first stages of coat assembly. We observed five patterns surrounding the OFM that were distinct in appearance and turned out to be three-dimensional objects that we named layers when they seemed organized or matrices when they appeared amorphous. In cells displaying a curved septum (3 stage-II$_E$ *ΔspoIVB* cells), a repeated pattern starts appearing at the surface of the forespore (Fig. 3a, cyan

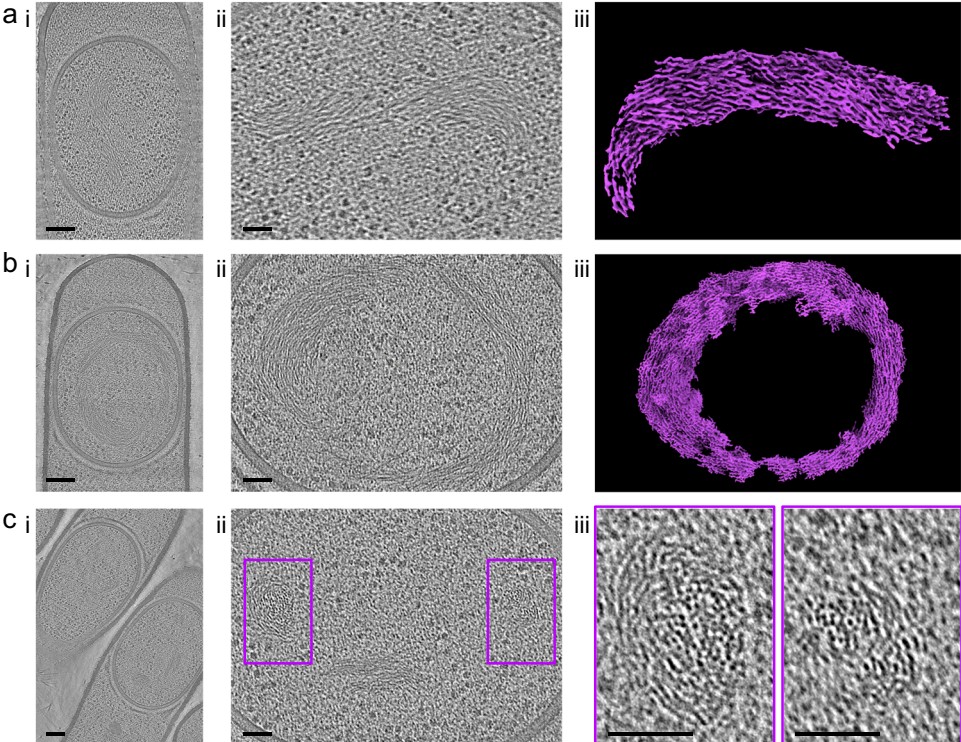

**Fig. 2 | DNA organization in the *B. subtilis* forespore. a, b** Full views (**i**, scale bars = 100 nm) and zooms (**ii**, scale bars = 50 nm) of an engulfed forespore harboring organized crescent- (**a**) or toroid-shaped (**b**) DNA structures. DNA segmentation (**iii**) suggests that it forms short filamentous structures that organize around a DNA-free region. **c** Full view (**i**, scale bars = 100 nm) and zooms (**ii** and violet insets, scale bars = 50 nm) of an engulfed forespore harboring two discrete DNA-rich regions. The images are representative of 2 independent experiments, with 18 (**a**), 5 (**b**) and 10 (**c**) cells displaying similar features.

beads). Its proximity to the OFM results in a crenelated aspect, which we have adopted to name this innermost layer. Although too faint to be unambiguously segmented, this crenelated layer covers larger regions around mid- (5 stage-II$_M$ $\Delta$spoIVB cells) and post-engulfment (15 stage-III $\Delta$spoIVB cells) forespores, rising to 12.4 nm ± 0.7 nm (*n* = 18) above the OFM (Fig. 3b–d, Supplementary Fig. 3a and Supplementary Table 2). The crenelated layer is embedded in a light matrix, which becomes more defined after engulfment, with an approximate thickness of 31.3 ± 2.4 nm (*n* = 24) (Fig. 3b–d and Supplementary Fig. 3a,b, cyan bracket). From mid- to post-engulfment stages, the light matrix gets surrounded by a dark matrix that displays a thickness of 17.8 ± 4.9 nm (*n* = 24) (Fig. 3b, c and Supplementary Fig. 3a,b, lime border or lime bracket). On top of this dense matrix, a bead-like pattern becomes visible halfway through engulfment (thickness of 8.3 ± 0.8 nm, Fig. 3b–d and Supplementary Fig. 3a, b, red beads). Its distance from the OFM is 55.1 ± 2.7 nm (*n* = 24). A second bead-like pattern forms 7.2 ± 0.9 nm (*n* = 24) above the first one (Fig. 3c, d and Supplementary Fig. 3a, b, orange beads). These two patterns adopt the curvature of the OFM, with a wavy appearance in some areas. They are visible on the mother-cell proximal side of all engulfed forespores, and on the mother-cell distal side in 9 out of 15 cells, probably because mother-cell distal coat layers assemble later than mother-cell proximal ones. Some $\Delta$spoIVB post-engulfment sporangia (7 out of 15) show a dark smooth pattern (thickness of 7.9 ± 1.0 nm, *n* = 24, Supplementary Table 2), which assembles above the bead-like patterns at the mother-cell proximal pole (Fig. 3d and Supplementary Fig. 3b, green line or green arrowhead). Segmentation of the different objects shows that all the coat patterns extend over all or part of the section (Fig. 3e). In particular, patches formed by the bead-like layers extend throughout the whole section whereas the dark smooth layer does not. In order to analyze the structure of the different coat layers, we examined the tomograms in planes orthogonal to the section (Fig. 3f). This analysis

shows that the bead-like layers are less dense than the dark smooth layer (Fig. 3f, compare panel v with panel vi), indicating that they probably involve different coat proteins.

**Contribution of morphogenetic proteins to nascent coat layers**
The assembly of the coat layers is initiated by morphogenetic proteins, among which SpoVM, SpoIVA, SpoVID, SafA and CotE have been the most studied[9]. Since SpoIVA polymerizes into filaments tethered to the OFM through their interaction with SpoVM[20,21], SpoVM and SpoIVA most likely form the crenelated layer adjacent to the OFM. CotE is so far the only other coat protein whose propensity to form filaments has been demonstrated, and it is required for the assembly of the outer coat[22,23]. We thus reasoned that CotE might contribute to the structured patterns observed by cryo-FIBM/ET at the OFM-distal position, or in other words to the bead-like or the dark smooth layers. A previous study had reported the encasement defect of a CotE-GFP fusion in the absence of SpoIIQ[18]. In $\Delta$spoIIQ sporangia observed by cryo-FIBM/ET, the dark smooth layer remains unilamellar while the bead-like layers accumulate (Fig. 4a). In addition, the bead-like patterns are not observed in $\Delta$cotE sporangia (*n* = 40 stage-III sporangia) whereas the other layers are present (Fig. 4b and Supplementary Fig. 3d-e). Altogether, these observations suggest that the bead-like layers are made of CotE polymers. Moreover, in the absence of CotE, the dark matrix sometimes extends along mother cell membrane regions that are close to the forespore (Fig. 4c). In $\Delta$spoIVB cells, such localization of the dark matrix is never observed. As the bead-like layers are not present in $\Delta$cotE sporangia, one might speculate that they are required to confine the dark matrix around the forespore.

In 6 $\Delta$cotE cells (out of 40 stage-III sporangia), the dark matrix appears directly capped by the dark smooth layer, most often in regions close to the mother cell cytoplasmic membrane (Fig. 4b). Analysis of the dark smooth layer in planes orthogonal to the lamella

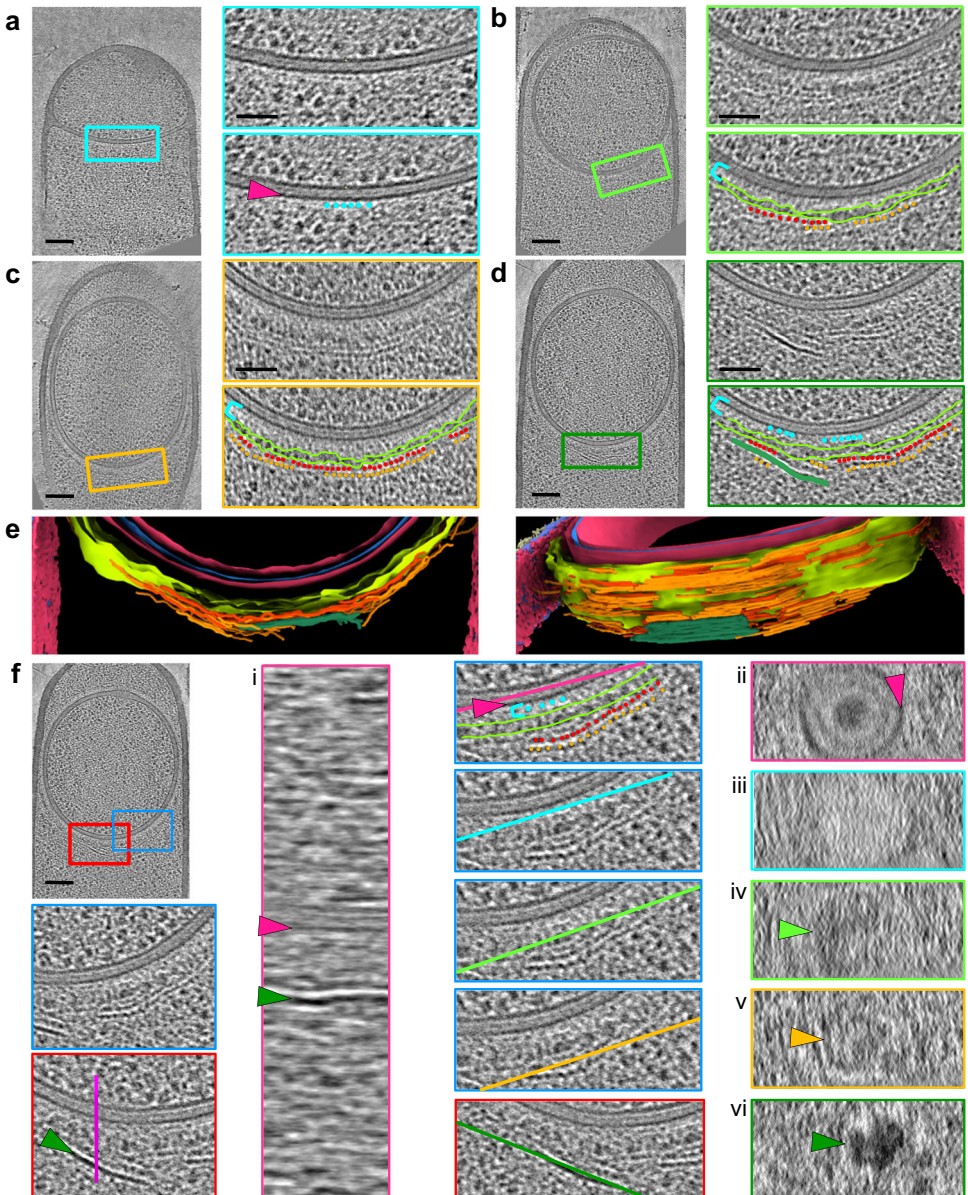

**Fig. 3 | Early coat assembly around the *B. subtilis* forespore. a–d** Slices through cryo-electron tomograms of *ΔspoIVB* forespores at different stages of development, shown in full view (scale bars = 100 nm) or as magnified views of specific regions (scale bars = 50 nm). The insets box the zoomed area. Five different regions can be distinguished above the OFM (magenta arrowhead): a crenelated layer (cyan beads), a light matrix (cyan bracket), a dark matrix (delineated in lime), two bead-like layers (red and orange beads) and a dark smooth layer (green line). **e** Two views of the segmented coat layers. The color code is identical to that described in Fig. 1.

**f** Analysis of the 3D organization of early coat layers by observing their aspect in planes orthogonal to the section. The blue and red insets box the analyzed regions; the violet, magenta, cyan, lime, orange and green lines indicate the orientation of the plane used to generate panels **i–vi**. The symbol legend is the same as in **a–d**; the lime, orange and green arrowheads point to the patches formed by the dark matrix, the bead-like layers and the dark smooth layer, respectively. Scale bar = 100 nm. The images are representative of 2 independent experiments, with 6 (**a**), 5 (**b**), 15 (**c**), 7 (**d**) and 6 (**f**) cells displaying similar features.

shows that it forms patches (Fig. 4diii), as observed in the *ΔspoIVB* strain (Fig. 3fvi). Intriguingly in one *ΔcotE* sporangium, the dark smooth layer detaches from the dark matrix, with its forespore-proximal face forming a comb-like structure (Fig. 4e). The comb teeth are not visible through the whole section volume, suggesting that they are not formed by a domain of the protein that makes the dark smooth layer, but rather by a protein partner.

Given the low throughput of cryo-FIBM/ET, we further investigated the composition of the coat layers by performing TEM on sections of resin-embedded bacteria, stained with a contrasting agent. In order to preserve the native state of the sporangia ultrastructures, the samples were vitrified by high-pressure freezing before freeze-substitution and resin embedding. This method proved to be

complementary to cryo-FIBM/ET since it revealed seven regions of different staining properties and/or appearance. Proximal to the OFM, the *ΔspoIVB* strain shows a sandwich of four amorphous regions: a thin light matrix ($3.2 \pm 0.2$ nm thick, $n = 6$), a thin dark matrix ($8.9 \pm 0.8$ nm thick, $n = 6$), a thick light matrix ($17.8 \pm 3.7$ nm thick, $n = 6$) and finally a thick dark matrix ($22.5 \pm 4.5$ nm thick, $n = 24$) (Fig. 5a and Supplementary Table 3). The aspect, position and thickness of the two thick matrices suggest that they might correspond to the light and dark matrices observed by cryo-FIBM/ET, whereas the cumulated thickness of the two thin matrices matches that of the cryo-FIBM/ET crenelated layer (about 12 nm). Interestingly, the thin dark matrix is not observed in *ΔspoVID* sporangia and the thick dark matrix forms additional strata (Fig. 5b). Altogether, the architectural defects observed in the absence

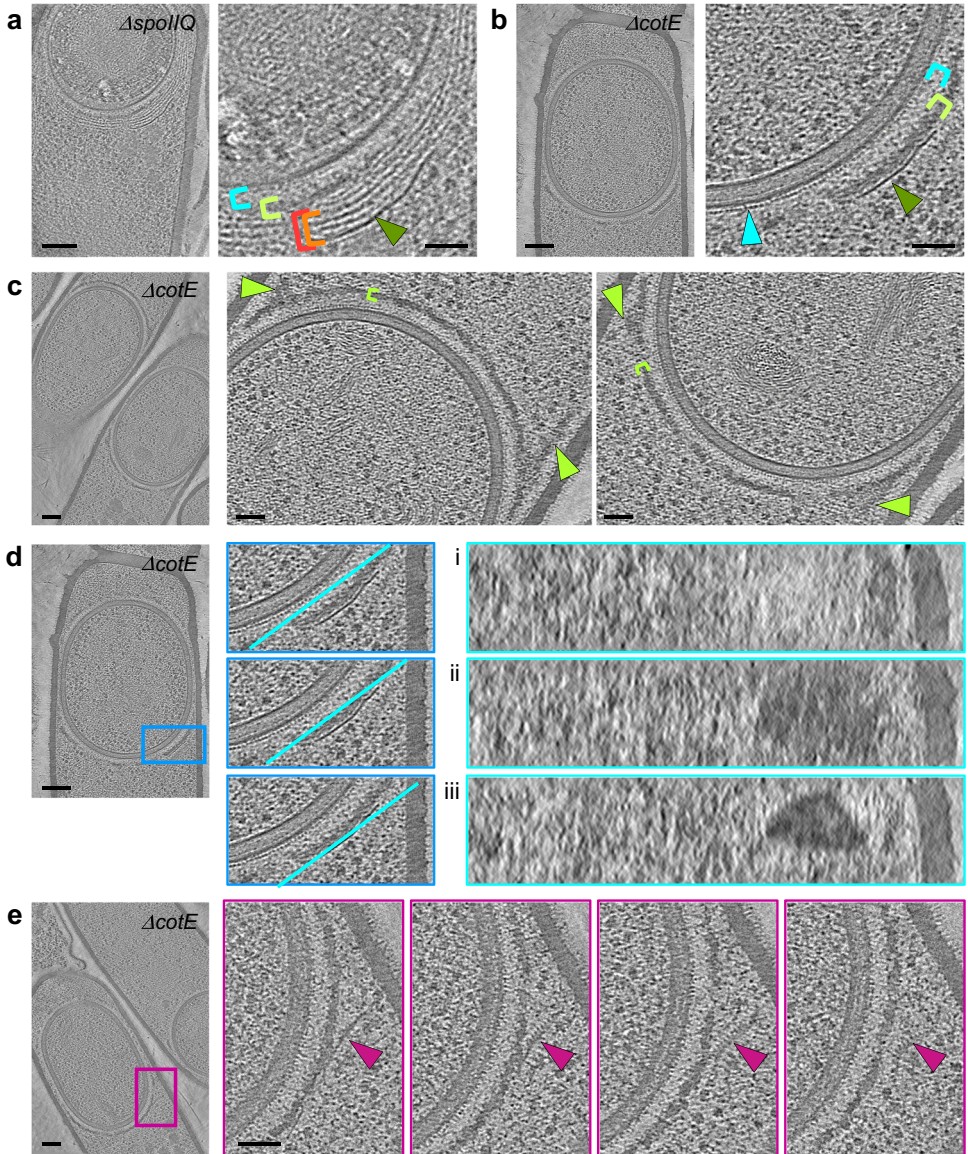

**Fig. 4 | Impaired early coat assembly in the absence of SpoIIQ or CotE. a–e** Slices through cryo-electron tomograms of *ΔspoIIQ* or *ΔcotE* forespores, shown in full view (scale bars = 100 nm) or as magnified views of specific regions (scale bars = 50 nm). **a** The bead-like patterns (red and orange brackets) accumulate in *ΔspoIIQ* sporangia. **b–c** In the absence of CotE, the crenelated layer (cyan arrowhead), the light (cyan brackets) and dark (lime bracket) matrices and the dark smooth layer (green arrowhead) are visible but the bead-like patterns are absent. In addition, the dark smooth layer localizes right above the dark matrix (**b**), which abnormally extends toward the mother cell (**c**, lime arrowheads). **d** Analysis of the 3D organization of the coat layers by observing their aspect in planes orthogonal to the section. The blue inset boxes the analyzed region; the cyan lines indicate the orientation of the plane used to generate panels **i–iii**. **e** Slices through a cryo-electron tomogram of a *ΔcotE* forespore shown in full view and as magnified views with incremental depth (boxed in violet). A comb-like structure protrudes from the dark smooth layer in regions where it detaches from the dark matrix. The images are representative of 2 independent experiments, with 2 (**a**), 6 (**b**), 3 (**c**), 6 (**d**) and 1 (**e**) cells displaying similar features.

of SpoVID, indicate that this morphogenetic protein contributes to the thin dark and the thick dark matrices.

Above the thick dark matrix in *ΔspoIVB* sporangia, we observed two thin structured layers (Fig. 5a) whose position (distance to the OFM = 56.3 ± 6.3 nm, *n* = 13) matches that of the bead-like patterns observed by cryo-FIBM/ET (distance to the OFM = 55.1 ± 2.7 nm, *n* = 24). Like the bead-like patterns in cryo-tomograms, the thin structured layers disappear from TEM images in the absence of CotE (Fig. 5c), suggesting that they are made of CotE polymers.

Finally, at the distal position relative to the OFM, a thicker and dark structured layer (Fig. 5a) likely corresponds to the dark smooth layer observed in tomograms. Indeed, similar to cryo-FIBM/ET observations, the dark structured layer in *ΔcotE* sporangia forms right above the thick dark matrix (Fig. 5c). In TEM images of some *ΔcotE* sporangia, we furthermore observe the abnormal localization of the thick dark matrix, which extends toward the mother cell (Fig. 5c). Intriguingly in a *ΔsafA* mutant, the thick dark matrix is either absent or mislocalized as a large aggregate on top of the dark structured layer (Fig. 5d). This indicates that the thick dark matrix cannot assemble properly in the absence of SafA. Finally in the absence of SpoIVA, which is required to tether the coat to the OFM[9], all the nascent coat layers are mislocalized (Fig. 5e).

## Discussion
Resistance of spore DNA to various stresses, such as UV irradiation, desiccation and wet heat, requires its packing by SASP proteins[24].

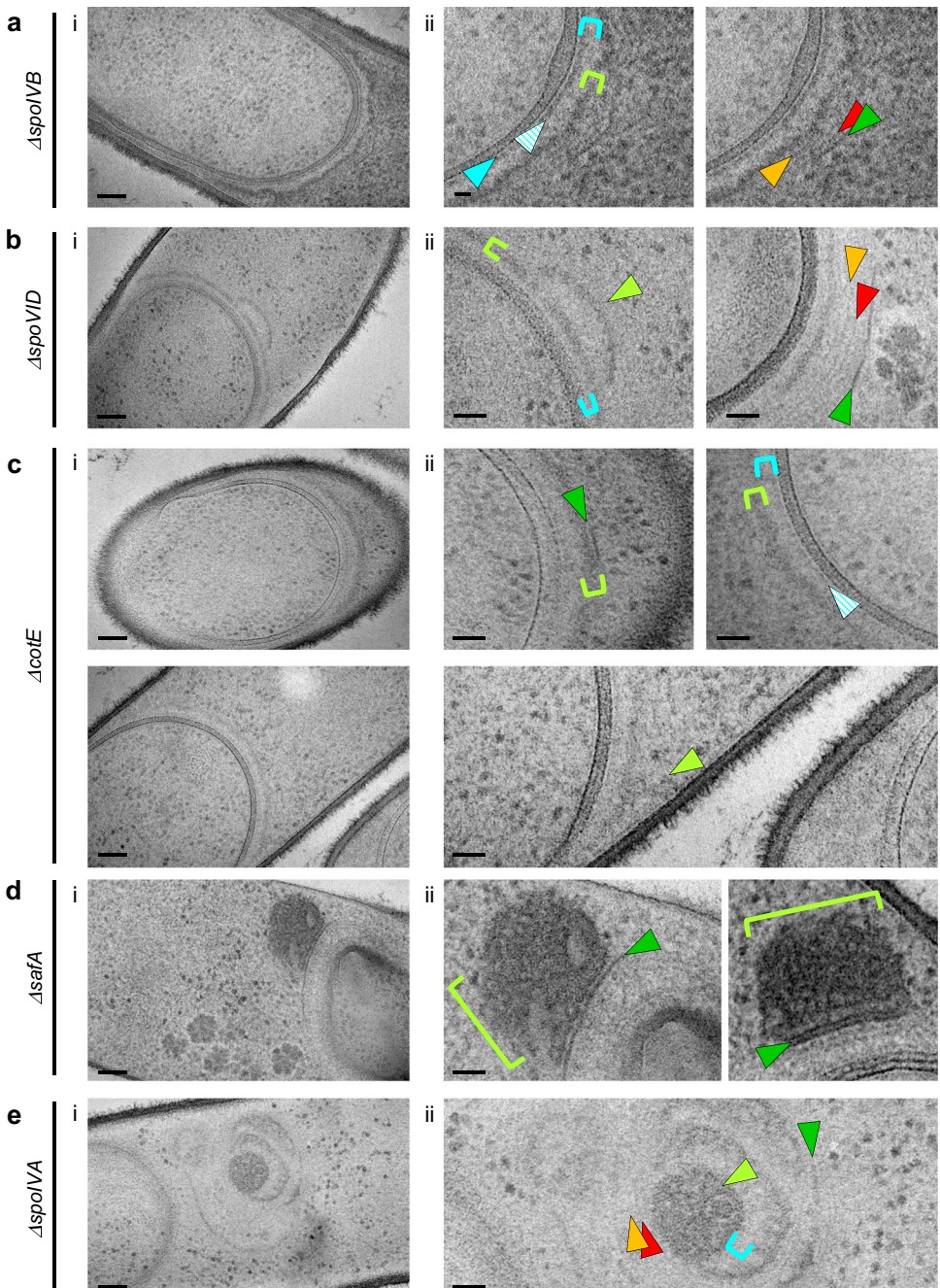

**Fig. 5 | Role of morphogenetic proteins in early coat assembly. a–e** Full views (**i**, scale bars = 100 nm) and zooms (**ii**, scale bars = 20 nm) of TEM micrographs collected on resin sections of *B. subtilis* sporangia. In the *ΔspoIVB* strain (**a**), different early coat layers can be distinguished above the OFM: a thin light matrix (cyan arrowhead), a thin dark matrix (cyan hatched arrowhead), a thick light matrix (cyan bracket), a thick dark matrix (lime bracket), two thin structured layers (red and orange arrowheads) and finally a thick structured layer (green arrowhead). In *ΔspoVID* (**b**), *ΔcotE* (**c**), *ΔsafA* (**d**) and *ΔspoIVA* (**e**) sporangia, the coat layer organization shows various architectural defects described in the text. The images are representative of 2 independent experiments, with 4 (**a**), 5 (**b**), 3 (**c**), 3 (**d**) and 10 (**e**) cells displaying similar features.

When DNA is incubated with SASP in vitro, it forms 5.1 nm-thick helical filaments that assemble into toroidal bundles[25]. Consistent with this, a toroidal structure of the chromosome was observed in vivo by TEM in resin sections of mature spores and by fluorescence microscopy in germinating spores[25,26]. In addition to the toroidal organization of the spore chromatin, a SASP-dependent crystalline arrangement was observed by CEMOVIS (cryo-electron microscopy of vitreous sections), from which the authors deduced the ordered packing of 5.5 nm-thick SASP-DNA bundles[27]. Since SASP genes are under the control of σG, which becomes active at the end

of engulfment, DNA bundles should be observed from this stage[24,28]. In our study, tomogram reconstruction and segmentation shows that indeed, just after engulfment, the DNA in the forespore exhibits a fibrillary structure, with bundles displaying a diameter of about 5.5 nm. The DNA ultrastructure is moreover consistent with a toroidal organization, which therefore appears to precede crystal packing. Ring-like organization of chromatin, which favors DNA protection and repair, is also observed in cells with low energy pools, such as steady-state *Escherichia coli*, or in bacterial species that are particularly adapted to DNA-damaging stress conditions,

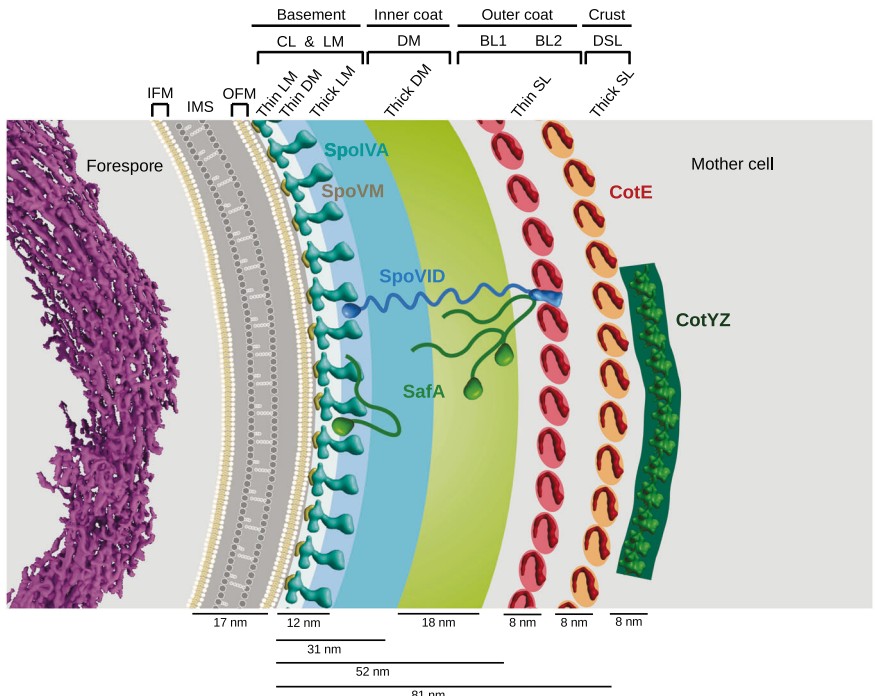

**Fig. 6 | Model of transient structures formed by the DNA and the coat, two cellular components involved in spore resistance.** In the cytoplasm of the forespore, the DNA (in violet) harbors a toroidal fibrillar conformation. A thin layer of PG is represented in the intermembrane space (IMS), delineated by the inner forespore membrane (IFM) and the outer forespore membrane (OFM). Morphogenetic proteins contributing to the seven coat regions evidenced by cryo-FIBM/ET and TEM are represented based on AlphaFold predictions. The crenelated layer (CL) observed by cryo-FIBM/ET likely corresponds to the thin light matrix (thin LM, in light blue) and thin dark matrix (thin DM, in blue) observed by TEM. Together with the thick light matrix (thick LM, in cyan), they would constitute the basement layer. In this model, the thick dark matrix (thick DM, in lime) would correspond to the inner coat, the CotE-dependent bead-like layers (BL, in red and orange) to the outer coat, and the dark smooth layer (DSL, in dark green) to the crust. The nascent coat layers are represented at the scale of the experimental measurements.

such as radiation-resistant *Deinococcus radiodurans*[12]. In starved *E. coli*, the toroidal chromatin additionally exhibits a crystalline order, which provides structural protection to the DNA. Altogether, these observations therefore support a stepwise protection mechanism for the spore DNA. In its fibrillar toroidal form (Fig. 6), it would benefit from a first level of protection by physical sequestration, which would still allow gene expression. Transition to a crystalline form would then provide a higher level of protection to the chromosome, more stable in the long term but genetically inert.

The envelope is a crucial region of the spore: it is not only the site of many communication processes between the mother cell and forespore, but it also hosts the different protective layers that make the spore a virtually indestructible microorganism. Despite the thinness of our FIBM/SEM sections, we could not clearly distinguish macromolecular complexes localized at the mother cell and forespore interface, such as the SpoIIIE DNA pump or the SpoIIIA-SpoIIQ complex. This is certainly due to the high molecular crowding of the IMS, as evidenced by its dark visual appearance in our tomograms. Further structural characterization of this region will thus require working with yet-to-be identified mutant strains harboring a lower molecular density in the IMS. By contrast, at the surface of the OFM, we observed with high definition the first stages of coat assembly, which shows seven main areas (Fig. 6). In cryo-electron tomograms, the innermost layer is a repeated motif that is likely formed by SpoIVA filaments, which are anchored in the OFM through interaction with SpoVM[20,29]. On top of this layer, our tomograms show two thick amorphous matrices of different electron density, suggestive of protein assemblies that do not form any ordered pattern. In previous TEM studies of resin sections, a single amorphous region had been observed and proposed to be a scaffold region for CotE-independent coat components[30]. Two layers with distinct molecular properties assemble independently of CotE: the basement layer (SpoIVA-dependent assembly), and the inner coat

(SpoIVA- and SafA-dependent assembly)[9]. The basement layer contains about half as much proteins as the inner coat. In our tomograms, the light matrix thus likely corresponds to the basement layer and the dark matrix to the inner coat (Fig. 6). At the post-engulfment stage, the morphogenetic basement proteins SpoVM, SpoIVA and SpoVID, as well as the morphogenetic inner coat protein SafA are present around the forespore[9,18,31]. Based on the reported interactions between these proteins, the basement layer would include SpoVM-SpoIVA filaments (observed as a crenelated layer in cryo-FIBM/ET), the C-terminal LysM-containing region of SpoVID, but also a yet-to-be-defined portion of SafA (Fig. 6)[31–33]. In TEM images, the region in which SpoIVA and SpoVID interact might correspond to the thin dark matrix since it disappears in *ΔspoVID* sporangia. Furthermore, since the N-terminal SPOCS (SpoVID-CotE-SipL) domain of SpoVID interacts with CotE[34], SpoVID would thus extend throughout the basement and inner coat layers.

The inner coat requires the morphogenetic protein SafA, which we show to be crucial for the position and structure of the thick dark matrix in our TEM studies of resin sections. The thick dark matrix thus likely corresponds to the inner coat layer. Interestingly, the assembly of the thick dark matrix is also impaired in the absence of SpoVID. The inner coat thus likely involves SafA and SpoVID. Consistent with this, the N-terminal region of SafA interacts with the N-terminal SPOCS domain of SpoVID[35,36]. The localization of SafA is however ambiguous since its N-terminal LysM domain also binds cortex-associated molecules, which are closer to the basement layer than to the inner coat[37]. In *B. subtilis* sporangia, SafA is present under three forms that self-interact[35,38]. The complex interaction network established between SafA and different components of the spore envelope might thus involve transient binding events, topologies that would be specific to the different forms of SafA, or a homo-oligomer spanning the basement and inner coat layers.

In our tomograms, CotE polymers would form the bead-like patterns, which top the dark matrix (Fig. 6). Indeed, the bead-like layers are not observed in the absence of CotE and they accumulate on the mother-cell proximal pole in the absence of SpoIIQ, as previously shown for a CotE-GFP fusion[18]. In this model, the position of the dark smooth layer, which localizes above the bead-like layers, suggests that it is either made of outer coat or crust proteins. Since the assembly of the outer coat and the crust is CotE-dependent and the dark smooth layer still forms in ∆cotE cells, it is likely formed by polymers of another morphogenetic protein. To our knowledge, above CotE in the hierarchy of assembly, candidates include homo-oligomers of CotY and hetero-oligomers of CotY-CotZ[39].

In conclusion, while further studies using mutant strains are needed to understand how DNA organizes in the spore and to fully describe the composition and formation of the various coat layers, this work lays solid ultrastructure foundations for the dissection of mechanisms underlying spore resistance.

## Methods

### Bacterial strains and culture conditions
All *B. subtilis* strains were derived from the prototrophic strain 168 and are listed in Supplementary Table 1. *B. subtilis* cells were grown in CH medium at 37 °C to OD$_{600nm}$ ˜ 0.3–0.5. Sporulation was then induced by resuspension in sporulation medium at 37 °C, according to the method of Sterlini-Mandelstam[40].

### Preparation of the bacterial cell layer for cryo-FIBM/SEM
Three and a half µL of concentrated *B. subtilis* cells (OD$_{600nm}$ ˜ 20) were deposited onto glow-discharged holey carbon coated QUANTIFOIL R 3.5/1 200 mesh copper grids. The cells were frozen using a Vitrobot Mark IV (Thermo Fischer Scientific, TFS). Back blotting was performed with a grade-595 filter paper (Ted Pella Inc.) on the backside of the grid, and a homemade 3D printed plastic pad on the cell side (polymer Flexfill 98 A). Blotting parameters were optimized to facilitate further milling: 22 °C, 100% humidity (stopped during process), wait time: 5 sec, blot time: 5 sec, blot force: 5, blotting cycle x1. The grids were then quickly vitrified in milky liquid ethane. The grids were subsequently clipped onto FIB autogrids (TFS). This procedure resulted in perfectly preserved grids covered with a uniform layer of bacterial cells (Supplementary Fig. 4a). The grid bars were visible and all grid squares were exploitable. The cell layer was thin enough for fast milling of 10–20 µm-deep lamellae.

### Cryo-FIBM/SEM workflow
Cryo-FIBM/SEM was performed on a Versa 3D Dual Beam (TFS) or a Crossbeam 550 (Carl Zeiss Microsystems), two scanning electron microscopes combined with a gallium beam column for serial FIB milling and imaging. A PP3010 cryo-preparation system (Quorum), installed on both microscopes allowed safe transfer of the vitrified samples, sputter coating and micro-manufacturing at cryogenic temperature.

Inside the FIB chamber, an integrated gas injection system (GIS) was used to deposit an organometallic platinum layer in order to protect the specimen surface, prevent charging during milling and limit curtaining artifacts. The GIS system was precooled to 28 °C (capillary and reservoir switched off). The stage was tilted at 35 °, then the GIS needle was introduced 3-4 mm above the sample, and platinum was deposited for 2 min. Shorter platinum deposition time resulted in radiation damage by the FIB while longer deposition time led to bent lamellae, making polishing difficult. Sputtering in the Quorum preparation chamber was occasionally performed to reduce charging during imaging but it did not significantly affect the milling procedure.

At a stage tilt of 20–25°, a grid square containing a uniform layer of bacteria was brought into the coincidence point for serial FIB slicing and SEM imaging. 150–300 nm-thick lamellae (10-20 µm in width, 10-20 µm in depth) were then prepared using rectangular milling patterns and beam current of 1 nA to 0.37 nA for rough milling (cross section mode), 100 pA to 50 pA for fine milling (cross section mode) and 30 pA for polishing (loop mode). 20 × 0.2 µm² stress-relief trenches were made with a current of 1.5 nA[41]. To remove ice contamination or re-deposition that occurred during the milling procedure, a polishing step was performed[42]. To our experience, polishing should be performed <2–3 h before the lamellae are retrieved from the SEM chamber in order to limit re-deposition and fully benefit from the fine milling.

Using the Zeiss Crossbeam 550 A microscope, the initial rectangular patterns and the stress-relief trenches were designed using the available SmartFIB 1.17.0 software and programmed for automated milling. About 20 lamellae were automatically milled over 2 h and the best 8 to 12 lamellae were then manually polished over the day. In average, about 40 cells were sectioned in a 10 × 18 µm lamella. For the Zeiss crossbeam 550, SEM imaging was performed at 3 kV, 16 pA, with a dwell time of 800 ns (scanning speed 4). On the other hand, the Versa 3D microscope allowed direct monitoring using the FIB signal concomitant with milling, and SEM images were sparingly acquired (Supplementary Fig. 4b–d).

### Cryo-electron tomography
FIB lamellae were manually oriented perpendicular to the tilt axis before being loaded into the electron microscope. Tilt series were collected on a 300 kV TITAN KRIOS electron microscope (TFS) equipped with a post-column energy filter and K2 (∆spoIVB dataset) or K3 (∆cotE and ∆spoIIQ datasets) direct detection cameras (Ametek) at the CEITEC, Brno, Czech Republic, and at the ESRF CM01, Grenoble, France[43]. The sample was tilted from −50 to +50° with an increment of 2.5° (∆spoIVB and ∆spoIIQ datasets), or from −60 to +60° with an increment of 3.0° (∆cotE datasets), according to the dose symmetric Hagen scheme[44]. The 0° of the tilt series was defined by taking into account the intrinsic tilt of the lamellae caused by the milling angle. The tilt series were acquired using SerialEM 4.0[45] or Tomo 5.15 (TFS). To acquire tomograms of large area of the sporangium and observe ultrastructures with a strong contrast, the images were recorded at a defocus of −10 or −20 µm at nominal magnifications of 33,000 (pixel size of 4.4 Å) on the K2 camera or nominal magnification of 19,500 (pixel size of 4.5 Å) on the K3 camera, with a cumulative dose of ˜ 80-90 e-/Å².

### Image processing and segmentation
MotionCor2 1.6.4 was used to align the frames and dose-weight the tilt series according to the cumulative dose [46]. Tilt series alignment was performed with the IMOD 4.11.12 software package[47], using the patch tracking method as no fiducial marker was present in the sample. Tomogram reconstruction was performed with IMOD 4.11.12, using weighted back projection and SIRT-like filter for purposes of representation. AreTomo 1.3.4 was also used to align and reconstruct some ∆spoIIQ tomograms[48]. Automated segmentation was performed using a conventional neural network algorithm implemented in EMAN2 2.99[49]. Training sets were prepared for the mother cell membrane, PG, surface glycans and ribosomes. For each feature, the training set contained a few positive examples manually segmented in 2D, and an exhaustive representation of negative areas not containing the object of interest. After successful network training, the algorithm was applied to the entire tomographic volume.

For purpose of representation, forespore membranes, PG layer in the IMS, coat proteins, storage granules, the flagellum and the flowers on stems structures were segmented manually using the Drawing and Interpolator tools in IMOD 4.11.12[50]. The isosurface visualization of all cellular features was rendered using UCSF ChimeraX 1.6.1[51]. Small size particles corresponding to false positive were removed using the Hide dust tool in ChimeraX 1.6.1.

**High-pressure freezing, freeze substitution and ultramicrotomy**

*B. subtilis* cell pellets were dispensed on the 200-μm side of a 3-mm type A gold/copper platelet, covered with the flat side of a 3-mm type-B aluminum platelet (Leica Microsystems). The sample was vitrified by high-pressure freezing using an HPM100 system (Leica Microsystems) in which cells were subjected to a pressure of 210 MPa at −196 °C.

Following high pressure freezing, the vitrified pellets were freeze-substituted at − 90 °C for 80 h in acetone supplemented with 2 % $OsO_4$, the samples were warmed up slowly (2 °C/h) to −60 °C (AFS2; Leica Microsystems). After 8 to 12 h, the temperature was raised to −30 °C (2 °C/h), and then to 0 °C within 1 h to improve the osmium action and increase the membrane contrast. The samples were then cooled down to −30 °C within 30 min, before being rinsed 4 times in pure acetone. The samples were then infiltrated with gradually increasing concentrations of resin (Embed812, EMS) in acetone (1:2, 1:1, 2:1 [v/v]), during 2 h for each ratio, while raising the temperature to 20 °C. Pure resin was added at 20 °C. After polymerization at 60 °C for 48 h, 70-nm sections were obtained using an ultramicrotome UC7 (Leica Microsystems) and were collected on formvar carbon-coated 200-mesh copper grids (Agar Scientific). The thin sections were post-stained for 5 min in 2 % aqueous uranyl acetate, rinsed in water, incubated for 5 min in lead citrate and rinsed again. The samples were observed using a Tecnai G2 spirit BioTwin (FEI) microscope operating at 120 kV, at a magnification ranging from about 16,000 to 32,000 (pixel size of 0.56 and 0.28 nm), with an Orius SC1000B CCD camera (Gatan).

**Statistics and reproducibility**

Cryo-FIBM/ET and resin-TEM data collections of equivalent samples were performed twice independently and provided the same conclusions. The dimensions of the different objects present in our micrographs were manually measured with the IMOD tools (version 4.11.12). Statistical analysis was performed using the analysis of variance (ANOVA).

**Reporting summary**

Further information on research design is available in the Nature Portfolio Reporting Summary linked to this article.

## Data availability

The data that support this study are available from the corresponding authors upon request. The dataset that are necessary to interpret, verify and extend the research in the article are accessible through the Electron Microscopy Data Bank (EMDB) under accession codes EMD-19405 and EMD-19411, and through the Zenodo repository https://doi.org/10.5281/zenodo.8060225. The source data underlying Supplementary Tables 2 and 3 are provided as a Source Data file. Source data are provided with this paper.

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

## Acknowledgements

We thank members of the Morlot and Schoehn laboratories, and members of the microscopy platforms, for support, advice and encouragement, with special thanks to E. Neumann, D. Traore, O. Glushonkov and J.P. Kleman. IBS acknowledges integration into the Interdisciplinary Research Institute of Grenoble (IRIG, CEA). This work used the platforms of the Grenoble Instruct-ERIC center (ISBG; UAR 3518 CNRS-CEA-UGA-EMBL) within the Grenoble Partnership for Structural Biology (PSB), supported by FRISBI (ANR-10-INBS-0005-02) and GRAL, financed within the University Grenoble Alpes graduate school (Ecoles Universitaires de Recherche) CBH-EUR-GS (ANR-17-EURE-0003). The IBS/ISBG electron microscope facility is supported by the Auvergne-Rhône-Alpes Region, the Fondation Recherche Médicale (FRM), the fonds FEDER and the GIS-Infrastructures en Biologie Santé et Agronomie (IBISA). This work also benefited from access to beamline CM01 at the European Synchrotron Radiation Facility. We acknowledge cryo-electron microscopy and tomography core facility CEITEC MU of CIISB, Instruct-CZ Centre, supported by MEYS CR (LM2023042) and European Regional Development Fund-Project UP CIISB (No. CZ.02.1.01/0.0/0.0/18_046/0015974), Instruct-ERIC (PID1697 and PID17035), and iNEXT-Discovery (project number 871037) funded by the Horizon 2020 program of the European Commission. E.B. received funding from GRAL, a program from the Chemistry Biology Health (CBH) Graduate School of University Grenoble Alpes (ANR-17-EURE-0003), and C.D.A.R. from the Australian Research Council (grant DP190100793).

## Author contributions

E.B., C.D.A.R., C. Moriscot. and C. Morlot. designed research. E.B., B.G., G.E., H.C., J.N., G.S. and C. Moriscot. performed experiments. E.B. and C. Morlot. analyzed data. P.H.J., J.M. and G.S. provided support for FIBM/SEM and electron microscope access and training. E.B. and C. Morlot. wrote the manuscript, with input from all authors; E.B., B.G., J.N., C.D.A.R., G.S., C. Moriscot. and C. Morlot. revised the manuscript.

## Competing interests

The authors declare no competing interests.
