## [Peer Review File · Nature Communications]

Ultrastructure of macromolecular assemblies contributing to bacterial spore resistance revealed by in situ cryo-electron tomographyReviewer #1 (Remarks to the Author):

The study by Bauda et al. investigates the ultrastructure of sporulating *Bacillus subtilis* using a combination of cryo-FIB milling and TEM to assess the packaging of the DNA-SASP complex as well as the endospore coat proteins. Imaging of mutants deficient in major coat assembly proteins and the late sporulation regulation factor SpoIVB informed the possible structural roles of these spore components in the polymerization of the coat layers and cortex synthesis in situ. The authors generated beautiful tomographic reconstructions and I appreciated the additional information about the challenges they faced during lamellae production, as cryo-FIB milling is still a nascent technique.

Unfortunately, I struggle with seeing the novelty of this study. Spore coat assembly in *B. subtilis* has been examined using other imaging modalities and the role of SpoIVD, CotE, and SafA to spore coat assembly have been well-established. Additionally, while I appreciate the use of mutants to arrest spore coat assembly at specific stages, the study lacked a wild-type (WT) reference. It is difficult to rely on conclusions inferred from analysis of deletion mutants, such as thickness of coat layers and presence or absence of structural features, without a baseline.

Generally, the manuscript is not easy to follow and at times incomprehensible due to the use of overcomplicated vocabulary, such as "crenelated layer", "amorphous aspect" (Lines 190-191), "aberrant...structures" (Line 250). Perhaps, these phrases should be simplified or defined more clearly. In addition, the figures were quite busy and some revision would assist in demonstrating the authors' findings more clearly. Throughout the manuscript, it would be beneficial to refer to the colour of a feature of interest in a particular figure in the text to improve readability.

-The introduction could benefit from describing the roles of the investigated genes during sporulation. The text does not elaborate why these specific genes were chosen or what their products do for the spore maturation/integrity. It would also be helpful to define the term "sporangium".

-The terms used to define the various layers, such as "thick/thin" and "dark/light" matrix or lines are difficult to follow because all the layers appear very similar. For example, the "crenelated layer" is referred to as such but it resembles the "bead-like" layers in Fig. 1.

-It could be beneficial to include 3D modelling and/or subtomogram averaging of the suspect proteins to assess how they fit within their respective layers that were observed with imaging. Without high-resolution characterization of protein interactions, the novelty of these findings is lost on me, given that more detailed models for *Bacillus* spore coat assembly have been proposed before (<https://doi.org/10.1038/nrmicro2921>).

-The analysis of the toroidal bundles is very exciting, however, does not fit within the general theme of the study. Altogether, this observation should be part of separate manuscript with additional in vitro crystallization experiments to support the in situ observations.

Specific comments:

- 1) Lines 76-97: It would be helpful to have references to specific panels in Fig. S1A for each stage of sporulation.
- 2) Line 114: Authors should use cryo-FIB in place of cryo-FIBM to keep in line with the accepted nomenclature.
- 3) Line 134: Stage IV should be included in Fig. S1A to make the manuscript easier to follow.
- 4) Figure 1 is quite busy. Ribosomes and storage granules are not discussed in the text, so I would suggest removing these features from the segmentation in Fig. 1A. Additionally, increasing the size of the tomographic slice and including feature labels directly on the image would improve the ease of feature identification. In Fig. 1B-F, the use of blue staples and arrowheads to highlight the same feature (PG) and then using blue arrowheads for a different feature (thin lines within the PG) in Fig.

1E-F, is confusing. Additionally, the use of the hatched magenta arrowhead to indicate the mother cell membrane in tomographic slices (Fig. 1B and E) but also solid magenta in segmentations (Fig. 1A-C), is inconsistent. A density profile across the cell envelope would convey information about the thickness and spacing of these features.

5) Line 149: Was mother cell envelope bulging only observed in ~50% of the cells for each mutant? What did the rest of the cells look like and how does this relate to WT numbers?

6) Lines 150-153: Not sure what these two sentences are conveying. What exactly are "scars" in the context of engulfment and how are these observations significant?

7) Lines 166-169: The claim that the dark lines are nascent glycan strands that are either being incorporated into PG or still attached to CM doesn't make sense given that these two lines are both seen inside the mother cell membrane in Fig 1E.

8) Figure 3: It would be beneficial to have a box on the tomographic slice on the left side of Figs. 3A-D to indicate where the zoomed in region on the right is. The use of written labels to identify features in the top right panel of Fig. 3A-D would make it easier to follow. In Fig. 3B-D, the light matrix (teal staple) and the crenelated layer (teal dots) should use different colours to indicate that they are different features. The term "bracket" rather than "staple" is more appropriate. "Lemon" should be changed to light green (or lime) in the figure caption to more accurately describe the colour. I have difficulty seeing the patches by bead-like layers and the dark smooth layer in the orthogonal planes.

9) Lines 184-185: It became difficult to follow and identify these layers in the figures. References to the colour of these features in Fig. 3 would be appreciated throughout this section.

10) Line 205: Plans should be corrected to planes.

11) Line 300: the lack of macromolecular complexes in the data is likely due to the high defocus used for collecting the tomograms, not the high molecular crowding of the cytoplasm. It is likely a typo but if not, a justification of why -10 to -20 micron defocus was used for tomography should be included in the M&M section.

12) Figure 5: I liked the clearly-labeled mutants on the left of the TEM images in Figure 5, and feel that the other figures could benefit from having this information more clearly conveyed on the figure. Density profiles across the spore wall and coat would help communicate information about the spacing and thickness of layers that are present and help highlight absent layers.

13) Line 313: "Twice as less" could be revised to "half".

14) Line 333: Sentence could be revised to: "... closer to the basement layer than to the inner coat".

15) Figure 6: Labels for SpoIVA, SpoVID and SafA are poorly legible.

16) Figure S1: Panels B-D not referenced in text.

Reviewer #2 (Remarks to the Author):

In this article, Bauda et al. report on the ultrastructural organization of *Bacillus subtilis* sporulating cells using techniques such as cryo-electron tomography (cryo-ET) and cryo-focused ion beam micromachining (cryo-FIBM). Although spore formation has been studied for decades using classical imaging approaches such as transmission electron microscopy (TEM) and fluorescence microscopy, revisiting this developmental process with these novel imaging approaches has considerable merit. One issue with TEM is that the process of preparing the sample for imaging (which includes slicing, fixing, and staining of the sample) is highly disruptive and can cause artefacts, whereas the traditional fluorescence microscopy approaches are limited in their resolution. Although this is not the first cryo-EM study of sporulation in *B. subtilis*, the previously published articles by Tocheva et al (2013) and Khanna et al (2019) focused on peptidoglycan and engulfment, or the cell division machinery (Khanna et al. 2021). They did not pay much attention to the most conspicuous structure of the spore, the multilayered spore coat. Yet, the spore coat is one of the most complex bacterial structures composed of at least 80 different proteins that are assembled in a highly regulated temporal and spatial pathway. Here, Bauda et al. analyzed these spore coat layers in exquisite detail using a collection of carefully selected spore coat mutants to determine the identity of the layers. What is remarkable in their observations is that they confirm previous models of spore coat assembly (for instance that the

assembly process begins very early during engulfment as suggested by fluorescence microscopy, but overlooked by TEM), while at the same time opening new research avenues. In addition, they present detailed images of the bacterial chromosome in the maturing spore. In general, the experiments are well done, and the images are spectacular. I only have minor suggestions.

Minor points:

- 1) I am unsure about the title "Ultrastructural details of resistance factors", especially the term "resistance factors", I don't think that the spore coat and nucleoid truly qualify as "factors", even though they are indeed structures (or multicomponent assemblies) that contribute to resistance.
- 2) I understand the logic of studying sporulation mutants (Δ spoIVB and Δ cotE) that limit production of coat proteins to the SigE-dependent ones (l. 134-136), thus reducing the complexity of the structures assembled and simplifying the interpretation of the observations, but there are a few questions that arise due to the choice of the mutants:
 - a) Why choose a spoIVB mutant instead of a sigK mutant, considering that spoIVB may have additional roles than just acting as a regulator of SigK activation?
 - b) cotE has two promoters, one dependent on SigE and the other dependent on SigK and it has been reported that the timing of cotE expression affects spore coat morphology (Costa et al. 2007). Is there a risk that the CotE localization and the structure of the outer coat will be impaired if SigK activity is prevented?
- 3) In Fig. 1A and l. 434 (methods), some prominent structures in the mother cell cytoplasm are referred to as "storage granules", but they are not discussed in the main text. Any insight into what they could be? Have they been observed before? Are they specific to the mother cell or could they also be present in vegetative cells?

Ultrastructure of macromolecular assemblies contributing to bacterial spore resistance revealed by in situ cryo-electron tomography

Elda Bauda, Benoit Gallet, Jana Moravcova, Gregory Effantin, Helena Chan, Jiri Novacek, Pierre-Henri Jouneau, Christopher D.A. Rodrigues, Guy Schoehn, Christine Moriscot and Cecile Morlot

Ref.: Nature Communications manuscript NCOMMS-23-42062-T

RESPONSE TO REVIEWERS'S COMMENTS

Reviewer #1

Reviewer #1 (Remarks to the Author):

The study by Bauda et al. investigates the ultrastructure of sporulating *Bacillus subtilis* using a combination of cryo-FIB milling and TEM to assess the packaging of the DNA-SASP complex as well as the endospore coat proteins. Imaging of mutants deficient in major coat assembly proteins and the late sporulation regulation factor SpoIVB informed the possible structural roles of these spore components in the polymerization of the coat layers and cortex synthesis in situ. The authors generated beautiful tomographic reconstructions and I appreciated the additional information about the challenges they faced during lamellae production, as cryo-FIB milling is still a nascent technique.

Unfortunately, I struggle with seeing the novelty of this study. Spore coat assembly in *B. subtilis* has been examined using other imaging modalities and the role of SpoIVD, CotE, and SafA to spore coat assembly have been well-established.

SpoVID, CotE and SafA have been extensively studied using microbial genetics and fluorescence microscopy, highlighting their role in the dynamics of coat formation (hierarchy of assembly, low-/medium-resolution localization and timing) (reviewed in McKenney et al., 2013; Driks and Eichenberger, 2016). **However, it is important to note that the structural characterization of the different coat layers remains virtually unexplored to this day, especially during early sporulation stages.** A few excellent studies employing transmission electron microscopy (TEM) of resin-embedded cell sections or atomic force microscopy have unraveled the existence of multiple layers of different density and organization in mature spores, and in some cases, evidenced the contribution of morphogenetic proteins in such assembly (also reviewed in McKenney et al., 2013; Driks and Eichenberger, 2016). Homo- and heteromeric interactions between coat proteins are known to play a key role in the formation of the highly organized coat layers, but the structural details of the coat architecture and the role of the morphogenetic proteins in determining this architecture have yet to be revealed at the molecular level.

In vitro reconstitution approaches are limited by the insolubility of many of the coat proteins produced in recombinant form and the complexity of the molecular interactions involved. The study of coat formation thus requires cellular approaches. As mentioned above, fluorescence microscopy and TEM have long been employed for in situ observation of the spore coat. However, the spatial resolution achieved by these methods remains relatively limited, and does not allow characterizing the fine architecture of the coat layers. The major methodological novelty in our study is the implementation of cryo-FIBM-tomography (cryo-FIBM/ET), which allows high spatial resolution but is, as pointed by the 1st reviewer, "still a nascent technique". E. Tocheva, G. Jensen,

E. Villa and K. Pogliano pioneered the use of cryo-FIBM/ET in *B. subtilis*, but they focused on other events of the sporulation process, such as storage granules, peptidoglycan modifications, engulfment or asymmetric cell division (Tocheva et al., 2013; Khanna et al. 2019, 2021).

Our cryo-FIBM/ET data provide **information in a near-native state, at unprecedented resolution and in three dimensions regarding the structure of the nascent coat layers**. Complemented by transmission electron microscopy, it also highlights the contribution of morphogenetic proteins in the assembly of the early coat. As pointed out by the 2nd reviewer, our study **(i)** confirms earlier models of coat deposition based on fluorescence microscopy experiments that could not be validated by TEM and thus remained speculative (Driks, 1999; McKenney et al., 2013; Driks and Eichenberger, 2016), and **(ii)** opens new research avenues since it unravels the formation of structured layers very early in the sporulation cycle. Finally, we believe that the correlation we are establishing between cryo-FIBM/ET and transmission electron microscopy descriptions will help the community integrate observations generated by either method.

Additionally, while I appreciate the use of mutants to arrest spore coat assembly at specific stages, the study lacked a wild-type (WT) reference. It is difficult to rely on conclusions inferred from analysis of deletion mutants, such as thickness of coat layers and presence or absence of structural features, without a baseline.

We agree with the reviewer that in the future, cryo-FIBM/ET studies will have to be performed in wild-type *B. subtilis*. However, due to highly limited access to cryo-FIBM/ET equipment, we cannot collect additional tomography data for this manuscript. Moreover, there are so many coat proteins (encoded by over 80 genes) that in this pioneering study, it seems wiser to describe the assembly of the coat in a strain that expresses only a subset of coat proteins, and therefore to use the *ΔspoIVB* mutant as a reference. In this strain, early expression of coat genes under the control of σE is maintained but expression of late coat genes under the control of σK is abolished. To our knowledge, no modification of the localization of the σE -dependent morphogenetic proteins has ever been reported when σK is inactive. Moreover, it seems unlikely that the absence of SpoIVB, which is produced in the forespore and acts in the intermembrane space, drastically affects the deposition of morphogenetic coat proteins at the surface of the forespore.

Regarding the *ΔcotE* mutant, the major observation by cryo-FIBM/ET is the absence of the bead-like layers, indicating that these structures are made of, or require, CotE. In the *ΔspoIIQ* mutant, the accumulation of the bead-like layers on the mother cell proximal pole of the forespore corroborates the encasement defect of a CotE-GFP fusion protein previously observed in the absence of SpoIIQ (McKenney and Eichenberger, 2012). The simplest interpretation of our cryo-FIBM/ET observations is thus that the bead-like layers are made of CotE polymers.

Generally, the manuscript is not easy to follow and at times incomprehensible due to the use of overcomplicated vocabulary, such as “crenelated layer”, “amorphous aspect” (Lines 190-191), “aberrant...structures” (Line 250). Perhaps, these phrases should be simplified or defined more clearly.

The words employed to name the different coat layers were intended to refer to their density and aspect. This is, in our opinion, the most objective way to describe them. In order to clarify some of the terms that we used, we have defined them better in the revised manuscript (lines 175-179, 244).

In addition, the figures were quite busy and some revision would assist in demonstrating the authors' findings more clearly.

Due to length constraints for the manuscript, we have organized the figures in the most balanced way in terms of information and size. We do not wish to place any part of the main figures into the supplementary information because they show important features that have never been reported before, and should therefore be easily accessible by remaining in the main manuscript.

Throughout the manuscript, it would be beneficial to refer to the colour of a feature of interest in a particular figure in the text to improve readability.

As mentioned above, the words employed to name the different coat layers were intended to refer to their density and aspect. This is the standard way to describe objects of interest in electron microscopy studies and, in our opinion, the most objective way to refer to the different coat layers. We have used a homogenous color code to delineate, segment the different objects of interest and display them in the working model of Fig. 6. To facilitate the association between terms used in the text and objects shown in figures, we have specified the color and symbol codes of the coat layers when first cited in the revised manuscript (lines 177-197).

The introduction could benefit from describing the roles of the investigated genes during sporulation. The text does not elaborate why these specific genes were chosen or what their products do for the spore maturation/integrity. It would also be helpful to define the term “sporangium”.

In the *Introduction*, we have added sections to define the term sporangium (line 62), summarize the sequential activation of sigma factors (lines 63-66), and presented the role of the morphogenetic proteins (SpoVM, SpoIVA, SpoVID, SafA and CotE) investigated in our study (lines 90-99).

The terms used to define the various layers, such as “thick/thin” and “dark/light” matrix or lines are difficult to follow because all the layers appear very similar. For example, the “crenelated layer” is referred to as such but it resembles the “bead-like” layers in Fig. 1.

As mentioned above, we have used standard terms of description in electron microscopy and we kindly ask to retain this nomenclature. In order to clarify some of the terms, including the “crenelated layer”, we have expanded their description in the revised manuscript (lines 175-197).

It could be beneficial to include 3D modelling and/or subtomogram averaging of the suspect proteins to assess how they fit within their respective layers that were observed with imaging. Without high-resolution characterization of protein interactions, the novelty of these findings is lost on me, given that more detailed models for Bacillus spore coat assembly have been proposed before (<https://doi.org/10.1038/nrmicro2921>).

In this review (McKenney et al., 2013; <https://doi.org/10.1038/nrmicro2921>), the authors provide an extended overview of coat assembly based on genetics, transmission electron microscopy and light microscopy data. They include a distance-weighted interaction map inferred from the interpolated contouring of conventional fluorescence microscopy images and genetic dependencies (McKenney et al., 2010). Although this was groundbreaking work in the field, the resolution of the localization data was limited to 20-30 nm, while our cryo-FIBM-tomograms allow the distinction of object separated by 5 nm or less. Our study further provides spatial information in 3D and unveils the cellular environment surrounding the objects of interest.

Subtomogram averaging (STA), which is used for 3D structure determination of macromolecular complexes, requires tilt series collected at low defocus (< 5 μm) and high magnification (pixel size < 2 \AA /pixel). Such tilt series generate high-resolution tomograms of small regions of the cell and subsequent STA requires picking of thousands of objects of particles to produce a 3D model. It is

also important to note that in 2022, the average resolution of 3D structures obtained by STA was about 20 Å (EMDB statistics). The tilt series acquired in our study (high defocus and low magnification) are not suited for STA but for ultrastructural characterization, as they allow visualizing large regions of the cell with high contrast. Acquisition of tilt series at low defocus/high magnification for subtomogram averaging is ongoing but is beyond the scope of the study presented in this manuscript.

The analysis of the toroidal bundles is very exciting, however, does not fit within the general theme of the study. Altogether, this observation should be part of separate manuscript with additional *in vitro* crystallization experiments to support the *in situ* observations.

In vitro structural characterization of DNA-SASPs complexes was already reported, showing the formation of 5.1 nm-thick filaments and toroidal bundles (Frenkiel-Krispin et al., 2004; Lee et al., 2008). *In vivo*, a SASP-dependent crystalline arrangement of the DNA was observed by CEMOVIS (cryo-electron microscopy of vitrous sections), from which the authors deduced the ordered packing of 5.5 nm-thick SASP-DNA bundles (Dittmann et al., 2015). **DNA toroids were however not observed in the work of Dittmann et al.** Our cryo-FIBM/ET data and segmentation analysis therefore make the link between these *in vitro* and *in vivo* observations, as we show that *B. subtilis* forespores harbor DNA toroids made of 5.5 nm-thick bundles. Although exciting, this observation is not sufficient for a separate manuscript, but it does seem worth sharing it with the community. Since DNA packing and coat formation are essential mechanisms for spore resistance, as stated in the title, we chose to gather them in the present manuscript.

Specific comments:

1) Lines 76-97: It would be helpful to have references to specific panels in Fig. S1A for each stage of sporulation.

These references have been added in the revised manuscript.

2) Line 114: Authors should use cryo-FIB in place of cryo-FIBM to keep in line with the accepted nomenclature.

The standard "cryo-FIB" abbreviation for "cryo-focused ion beam" does not include the micromachining procedure used to generate the lamellae. Therefore, we ask to keep the "cryo-FIBM" abbreviation that has been recently proposed to designate "cryo-focused ion beam micromachining" (Moravkova et al., 2021).

3) Line 134: Stage IV should be included in Fig. S1A to make the manuscript easier to follow.

We have added stages IV-VI to Fig. S1. For space constraints, we have moved panels B-E of Fig. S1 to Fig. S4 and modified the text accordingly.

4) Figure 1 is quite busy. Ribosomes and storage granules are not discussed in the text, so I would suggest removing these features from the segmentation in Fig. 1A.

As suggested by the reviewer, features that are not discussed in the main text (ribosomes, flagellum, unidentified surface element and storage granules) have been removed from the segmentation in Fig. 1A and 1B; they are now shown in Fig. S2 with annotations.

Additionally, increasing the size of the tomographic slice and including feature labels directly on the image would improve the ease of feature identification.

For consistency and aesthetics of the main figures, we prefer not to change the size of the tomographic slices and not to include feature labels on the images. However, to ease feature identification, we have added tomographic slices of bigger size and harboring labels in the revised Fig. S3.

In Fig. 1B-F, the use of blue staples and arrowheads to highlight the same feature (PG) and then using blue arrowheads for a different feature (thin lines within the PG) in Fig. 1E-F, is confusing.

Thanks for pointing this; we have replaced the blue arrow by a blue staple to annotate consistently the PG in Fig. 1B.

Additionally, the use of the hatched magenta arrowhead to indicate the mother cell membrane in tomographic slices (Fig. 1B and E) but also solid magenta in segmentations (Fig. 1A-C), is inconsistent.

We had chosen different types of filling to differentiate the mother cell membrane from the forespore membranes on the tomographic slices, but using a hatched filling for segmentation makes the image less clear. If the hatched magenta arrowhead is confusing, we therefore prefer to use solid magenta for all membrane symbols, as in the revised manuscript.

A density profile across the cell envelope would convey information about the thickness and spacing of these features.

We are not sure whether a density profile would be useful for our analyses. Indeed, it would only provide two-dimensional information on the thickness and spacing of the various elements of the cell envelope at a single position. In our study, we took advantage of the three-dimensional nature of our tomograms to measure spacing and thickness at different positions in the x, y and z directions. The mean values and standard deviations of these measurements are reported in the Table S2.

5) Line 149: Was mother cell envelope bulging only observed in ~50% of the cells for each mutant? What did the rest of the cells look like and how does this relate to WT numbers?

We did indeed observe bulging in 50% of cells for each mutant. As the thickness of cryo-FIBM sections is 150-200 nm, it is very likely that the cells in which we did not observe bulging were sectioned within a region that did not include these elements. We do not have any cryo-FIBM/ET data on a wild-type strain but we do see these bulges by TEM in resin sections of wild-type sporangia.

6) Lines 150-153: Not sure what these two sentences are conveying. What exactly are "scars" in the context of engulfment and how are these observations significant?

If the bulges of the mother cell envelope had encircled the sporangium along its short axis, they might have corresponded to the junction region between the vegetative PG and the septal PG synthesized during asymmetric division. We replaced the term "scars" by this definition and rephrased these two sentences to clarify our comment (lines 141-144). In terms of significance, it seems to us that this observation, although still mysterious, is worth to be mentioned because it has never been reported before.

7) Lines 166-169: The claim that the dark lines are nascent glycan strands that are either being incorporated into PG or still attached to CM doesn't make sense given that these two lines are both seen inside the mother cell membrane in Fig 1E.

In Fig. 1E the mother cell membrane is the thin, dense region indicated by the magenta arrowhead. The two dark lines on the other hand are embedded in the PG layer, whose thickness is delineated by the blue staple. The dark lines could thus well be glycan strands.

8) Figure 3: It would be beneficial to have a box on the tomographic slice on the left side of Figs. 3A-D to indicate where the zoomed in region on the right is.

We have added these boxes to the revised manuscript.

The use of written labels to identify features in the top right panel of Fig. 3A-D would make it easier to follow.

As mentioned above, for consistency and aesthetics, we prefer not to include feature labels on the main figures. However, to ease feature identification, we have added tomographic slices of bigger size and harboring labels in the revised Fig. S3.

In Fig. 3B-D, the light matrix (teal staple) and the crenelated layer (teal dots) should use different colours to indicate that they are different features.

Since the crenelated layer and the light matrix both belong to the basement layer, we prefer to use the same color (cyan, which also displays a good contrast with the tomography images). The fact that they are different features is indicated by the use of different symbols (dots for the crenelated layer, staple for the light matrix).

The term "bracket" rather than "staple" is more appropriate.

We have replaced the term "staple" by "bracket" throughout the revised manuscript.

"Lemon" should be changed to light green (or lime) in the figure caption to more accurately describe the colour.

We have replaced the term "lemon" by "lime" throughout the revised manuscript.

I have difficulty seeing the patches by bead-like layers and the dark smooth layer in the orthogonal planes.

In Fig. 3F, to ease the identification of the patches formed by the dark matrix, the bead-like layers and the dark smooth layer, we have pointed them with lime, orange and green arrowheads, respectively.

9) Lines 184-185: It became difficult to follow and identify these layers in the figures. References to the colour of these features in Fig. 3 would be appreciated throughout this section.

As mentioned above, to facilitate the association between terms used in the text and objects shown in figures, we have specified the color and symbol codes of the coat layers when first cited in the revised manuscript (lines 177-197).

10) Line 205: Plans should be corrected to planes.

The term "plan" has been replaced by "plane" throughout the revised manuscript.

11) Line 300: the lack of macromolecular complexes in the data is likely due to the high defocus used for collecting the tomograms, not the high molecular crowding of the cytoplasm. It is likely a typo but if not, a justification of why -10 to -20 micron defocus was used for tomography should be included in the M&M section.

As mentioned at the beginning of the *Results* section, we have indeed used - 10 to - 20 μm defocus values to visualize large area of the cell with high contrast. cryo-FIBM/ET data at lower defocus, which we are currently processing, also show a very dense intermembrane space. We have added a sentence in the M&M section to justify the use of high defocus values (lines 402-403).

12) Figure 5: I liked the clearly-labeled mutants on the left of the TEM images in Figure 5, and feel that the other figures could benefit from having this information more clearly conveyed on the figure. Density profiles across the spore wall and coat would help communicate information about the spacing and thickness of layers that are present and help highlight absent layers.

We have added the name of the mutants in the revised Fig. 4, which shows images obtained from two mutants (*AspoIIQ* and *AcotE*). Fig. 3 on the other hand only shows images of one mutant (*AspoIVB*), we have thus not added the mutant name to keep this figure labelling lighter.

Regarding the density profiles, as mentioned above, cryo-FIBM-ET data analysis would not benefit from such 2D analysis since we have exploited the 3D nature of our tomograms to measure spacing and thickness at different positions in the x, y and z directions (reported in Supplementary Table 2). For TEM data, the contrast is not sufficient to generate density profiles that would provide more information than visual inspection of the images.

13) Line 313: "Twice as less" could be revised to "half".

We have used "half" instead of "twice less" in the revised manuscript (line 333).

14) Line 333: Sentence could be revised to: "... closer to the basement layer than to the inner coat".

Thanks, we have made the revision.

15) Figure 6: Labels for SpoIVA, SpoVID and SafA are poorly legible.

We have increased the size of the labels for the protein names.

16) Figure S1: Panels B-D not referenced in text.

They are now referenced as Fig. S4B-D (line 391).

Reviewer #2 (Remarks to the Author):

In this article, Bauda et al. report on the ultrastructural organization of *Bacillus subtilis* sporulating cells using techniques such as cryo-electron tomography (cryo-ET) and cryo-focused ion beam micromachining (cryo-FIBM). Although spore formation has been studied for decades using classical imaging approaches such as transmission electron microscopy (TEM) and fluorescence microscopy, revisiting this developmental process with these novel imaging approaches has considerable merit. One issue with TEM is that the process of preparing the sample for imaging (which includes slicing, fixing, and staining of the sample) is highly disruptive and can cause artefacts, whereas the traditional fluorescence microscopy approaches are limited

in their resolution. Although this is not the first cryo-EM study of sporulation in *B. subtilis*, the previously published articles by Tocheva et al (2013) and Khanna et al (2019) focused on peptidoglycan and engulfment, or the cell division machinery (Khanna et al. 2021). They did not pay much attention to the most conspicuous structure of the spore, the multilayered spore coat. Yet, the spore coat is one of the most complex bacterial structures composed of at least 80 different proteins that are assembled in a highly regulated temporal and spatial pathway. Here, Bauda et al. analyzed these spore coat layers in exquisite detail using a collection of carefully selected spore coat mutants to determine the identity of the layers. What is remarkable in their observations is that they confirm previous models of spore coat assembly (for instance that the assembly process begins very early during engulfment as suggested by fluorescence microscopy, but overlooked by TEM), while at the same time opening new research avenues. In addition, they present detailed images of the bacterial chromosome in the maturing spore. In general, the experiments are well done, and the images are spectacular. I only have minor suggestions.

We thank the referee for her/his enthusiastic feedback on our manuscript and we provide below our answer to his/her comments.

Minor points:

1) I am unsure about the title “Ultrastructural details of resistance factors”, especially the term “resistance factors”, I don’t think that the spore coat and nucleoid truly qualify as “factors”, even though they are indeed structures (or multicomponent assemblies) that contribute to resistance.

We propose to change the title to "Ultrastructure of macromolecular assemblies contributing to bacterial spore resistance revealed by in situ cryo-electron tomography".

2) I understand the logic of studying sporulation mutants ($\Delta spoIVB$ and $\Delta cotE$) that limit production of coat proteins to the SigE-dependent ones (l. 134-136), thus reducing the complexity of the structures assembled and simplifying the interpretation of the observations, but there are a few questions that arise due to the choice of the mutants:

a) Why choose a *spoIVB* mutant instead of a *sigK* mutant, considering that *spoIVB* may have additional roles than just acting as a regulator of SigK activation?

We agree with the reviewer that performing experiments in a $\Delta sigK$ would have been more straightforward to limit the number of coat proteins to σE -dependent ones. In addition to activate σK , the other main function of SpoIVB is to degrade the large oligomeric complex SpoIIIA-SpoIIQ at the end of the engulfment process. To our knowledge, SpoIVB has never been shown to influence coat assembly. To be fully transparent, we thus used a $\Delta spoIVB$ mutant to give us chances to use our cryo-FIBM/ET data to investigate both coat assembly and preserve the SpoIIIA-SpoIIQ complex, which has never been studied by cryo-FIBM/ET so far. As mentioned earlier, cryo-FIBM/ET is a low throughput approach, partly due to our highly limited access to FIBM/SEM equipment and powerful electron microscopes. Therefore, we first investigated the $\Delta spoIVB$ strain because it would provide data that would both inform us on the architecture of the coat layers and of the SpoIIIA-SpoIIQ complex.

b) *cotE* has two promoters, one dependent on SigE and the other dependent on SigK and it has been reported that the timing of *cotE* expression affects spore coat morphology (Costa et al. 2007). Is there a risk that the CotE localization and the structure of the outer coat will be impaired if SigK activity is prevented?

The timing of *cotE* expression indeed affects the architecture of the spore coat (and more precisely outer coat deposition), but only when *cotE* is placed under the sole control of σK , or in other words

when *cotE* expression is delayed to post-engulfment (Costa et al., 2007). In a $\Delta spoIVB$ mutant, early expression of *cotE* under σE control is preserved and the structure of the coat should thus not be impacted. Moreover, our study focuses on early stages of coat assembly (T1.5 to T3), during which σK is not yet active (σK activation happens around T3.5 when sporulation is induced by resuspension). We thus only observe pre-outer coat layers formed by σE -dependent CotE.

3) In Fig. 1A and l. 434 (methods), some prominent structures in the mother cell cytoplasm are referred to as “storage granules”, but they are not discussed in the main text. Any insight into what they could be? Have they been observed before? Are they specific to the mother cell or could they also be present in vegetative cells?

The objects that we annotated "storage granules" have never been observed before by cryo-FIBM/ET but some TEM studies on resin sections have shown similar structures.

To our knowledge, three types of storage granules have been reported in bacteria: phosphate granules, polyhydroxybutyrate (PHB) granules, and polysaccharide granules. Phosphate granules typically exhibit a rounded shape, they are highly dense under the electron beam, and are rather isolated objects when observed by cryo-FIBM/ET in various bacterial species (Cell Ultrastructure Atlas, Jensen & Tocheva, <https://www.cellstructureatlas.org/4-9-archaeal-storage-granules.html>). In sporulating bacteria, they are observed in the forespore cytoplasm (Tocheva et al., 2013, doi:10.1128/JB.00712-13).

PHB is a biopolymer produced by certain *Bacillus* species when cultivated in the absence of essential nutrients (Hamdy et al., 2022, doi:10.1186/s12866-022-02593-z; dos Santos et al., 2017, doi:10.17230/ingciencia.13.26.10). Although it has never been observed by cryo-FIBM/ET, PHB granules typically exhibit an extremely round and light appearance under the electron beam in resin sections (Grage et al., 2017, doi:10.1186/s12934-017-0823-5).

Glycogen storage granules are often characterized as abundant and prickly structures, densely packed within a specific region of the cytoplasm, as observed in resin sections of *S. antibioticus* (Brana et al., 1986, doi:10.1099/00221287-132-5-1319). Extracted glycogen storage granules from *N. astroides* exhibit a flower-like structure with dimensions of 60-100 nm (Dipersio & Deal, 1974, doi:10.1099/00221287-83-2-349), similar to the structures observed in our *B. subtilis* sporangia.

Based on these observations, it appears that the flower-like objects observed by cryo-FIBM/ET in the cytoplasm of the mother cell are most likely polysaccharide storage granules. In our tomograms, they are never observed in the forespore but are also present in vegetative cells.

Reviewer #1 (Remarks to the Author):

The authors have addressed my main concerns to the best of their ability.

Reviewer #2 (Remarks to the Author):

On the whole, I feel that the authors have done an excellent job addressing the reviewers comments and I can only reiterate how impressed I am by the quality of the images that were collected. I am convinced that this paper will have a lasting impact on the community of spore researchers. My only regret is that the authors chose to remove the storage granules from the figures instead of mentioning them in the text. I feel that the sporulation community would have benefited from learning about these structures and I enjoyed reading the explanation provided by the authors in their answer to point 3 from Reviewer#2.